# Wolves, Crows, Spiders, and People: A Qualitative Study Yielding a Three-Layer Framework for Understanding Human–Wildlife Relations

Uta M. Jürgens [1,2,*], Paul M. W. Hackett [3,4], Marcel Hunziker [2] and Anthony Patt [1]

1    Swiss Federal Institute of Technology ETH, Department of Environmental System Science, Universitätstrasse 22, 8092 Zürich, Switzerland; anthony.patt@usys.ethz.ch
2    Swiss Federal Research Institute WSL, Social Science in Landscape Research, Zürcherstraße 111, 8903 Birmensdorf, Switzerland; marcel.hunziker@wsl.ch
3    Emerson College, School of Communication, Boston, MA 02116, USA; paul_hackett@emerson.edu
4    School of Health Sciences, University of Suffolk, Ipswich IP4 1QJ, UK
*    Correspondence: u.m.juergens@uta.info

**Abstract:** Human dimensions research has proposed a multitude of variables impacting the viability of wildlife populations. Extant approaches to systematizing these variables have mostly focused on human relations to only one animal species or taxon and are largely descriptive, rather than explanatory. In this study, we provide a three-layer framework for understanding people's responses to a variety of human–wildlife encounters. We conducted a comparative qualitative study, interviewing 20 stakeholders on one of three ecologically disparate model animals. Through thematic analysis, we identified person-specific, species-specific, and overarching factors whose interplay shapes people's reactions to encounters with wildlife. The person-specific factors, individual people's biographic backgrounds and life themes, fuel the polarization of stances towards wildlife. The species-specific factors, people's mental images of wild animals, explain the particular character of different human–wildlife relations. The overarching factors, fundamental questions regarding the place of humans in nature or motivations of control over animal agents, stir the intensity inherent in human encounters with wildlife. This three-layer framework amends existing proposals by providing a cohesive system and an in-depth portrayal of shared and specific factors and processes in various human–wildlife relations and by elucidating their interaction in influencing people's responses to encounters with wild animals.

**Keywords:** human dimensions; wildlife; human–wildlife conflict; human–wildlife coexistence; human–wildlife relations; qualitative research; wolves; value orientations; control

## 1. Introduction

Treves and Karanth stated, in the case of wolves, that "conservation depends on the sociopolitical landscape as much as the biological landscape." [1] (p. 1491). Wolves may be seen as emblematic examples for how intricate and polarizing human coexistence with wildlife in the cultural landscape may be, and for the fact that human attitudes, beliefs, and values constitute one of the chief impact factors in biodiversity restoration and conservation. Yet, this is likewise true for many other cases of human–wildlife interactions [2]. People's conceptions of wild animals inform their behavior towards them, particularly through wildlife management. Management, in turn, is an evolutive factor for the viability of wildlife populations [3]. Conversely, human attitudes towards wild animals responds to wildlife behavior [2,4,5]. Thus, human psychology and wildlife ecology co-create complex socio-ecological linkages [6–11].

The exploration of these linkages has grown over recent years. Still, human dimensions research has largely focused on the socio-economic facets and demographic variables

associated with diverging views of human interactions with a limited scope of species or groups of species, e.g., large predators or other charismatic wildlife (e.g., [12–16]). Few studies have taken explanatory, rather than descriptive, approaches [17], and if so, have presented one causal factor at the expense of the many further ones that exist. Studies comparatively assessing multiple cases of human–wildlife relations are particularly scarce [18]. In consequence, there are only a handful of approaches to systematically map and interrelate the various aspects contributing to human–wildlife relations and to identify potentially overarching factors [19]. While evidence for overarching factors and dynamics emerges [2,20], we lack an understanding of how they play out in human–wildlife interactions as diverse as Warli and Batswanan's appraisal of large felines [21,22], Sàmi and Eastern Germans' relations to wolves [23,24], or New York and Massachusetts residents' views on beavers [25]. We know even less about how the idea of overarching attitudinal dynamics may be reconciled with the polarization of humans' attitudes and with the stunning diversity of people's responses to wildlife ranging from stewardship to manifest intolerance [17]. Finally, it remains to be explained how all of the contributing factors interplay to create a person's reaction in a given encounter with a wild animal.

In this paper, we aim to fill these research gaps. First, we review extant summaries of the variables discussed in human dimensions research. We then present a straightforward three-layer framework based on the results of a comparative investigation that identify specific and overarching factors and processes in which both critical and favorable views of ecologically disparate wildlife are rooted.

### 1.1. Extant Synopses of the Human Dimensions in Wildlife Conservation and Management

A handful of authors have provided summaries of the factors that previous research identified as being relevant to human–wildlife interactions. In this section, we review extant synopses and thereby give an overview of the current state of knowledge on potential overarching dynamics in human relations to wildlife. Appendix A Table A1 shows how the variables named in these synopses and canonical concepts in human dimensions research correspond with one another and with the factors and processes that emerged from our empirical investigation (for a detailed discussion, see Section 3.3.3).

König et al. [26] have proposed a formal model for systematizing factors contributing to human–wildlife conflict and coexistence. They distinguish four organizational levels of management approaches and tools, ranging from international to regional to local. Distinguishing levels of scientific inquiry into human–wildlife relations, Manfredo and Dayer [27] define a micro-level—focusing on affect, cognition, and behavior of individual people—and a macro-level, focusing on societal and cultural aspects. Micro-level factors may be individual attitudes, norms, values, and "perceptions of control" (ibid., 319); macro-level factors are constituted by cultural characters, e.g., collective views of nature within a society.

Targeting both micro- and macro-level mechanisms that may be shared between the diverse forms that human–wildlife interactions take, Dickman [20] argues that attitudinal factors moderate the perception and management of virtually all human–wildlife conflicts. She proposes environmental and social risk factors that fuel people's disproportionate responses to challenges in human–wildlife coexistence: Environmental risk factors may be physical features of the environment, as well as the behavior of humans and wildlife. Social risk factors are constituted by human–human conflicts underlying alleged human–wildlife conflicts, e.g., urban–rural divide or wealth inequalities. Moreover, symbolism and cultural perspectives may stir and maintain conflict even when objective challenges are mitigated.

These and further variables are included by Bhatia et al. [17] in a collection of 55 proximate factors, from which they synthesize five ultimate factors that have been proposed to shape humans' multivariate responses to encounters with wildlife: value orientations, social interactions, resource dependence, perception of risk, and the nature of interaction with the animal.

An elaborate model has been developed by Kansky and Knight [19] and Kansky, Kidd, and Knight [28]. The former list factors that have been found to impact human responses to mammalian wildlife: costs and benefits, particularly those of an intangible nature such as negative or positive emotions; appraisals of ecosystem services or aesthetics; characteristics of the land such as ownership rights and landscape features; attitudes; prior experience with wildlife; demographic group, cohort, and socio-demographic variables; the "presence or absence, abundance, density, or the frequency" with which a wildlife species is "observed" (ibid., p. 97); people's knowledge about the species; mitigation measures taken; and the relation of stakeholders to institutions.

From this collection, Kansky et al. [28] constructed the wildlife tolerance model, by which they propose that acceptance of human coexistence with mammalian wildlife is contingent on two tiers of variables that contribute to perceived costs and benefits which, in turn, mediate tolerance towards wildlife. The major "outer" tier comprises of the tangible and intangible costs and benefits as well as positive and negative experience with and exposure to a species which directly drive tolerance. People's appraisal of costs and benefits, in turn, is influenced by eleven factors comprising the "inner" tier of the model: wildlife value orientation and general values, anthropomorphism, interest in animals, taxonomic bias, personal and social norms, trust in institutions, empathy, perceived behavioral control, and habit.

While the approaches of Kansky and colleagues accommodate a variety of factors discussed within the field of human dimensions research, they only focus on mammalian wildlife. Even for this relatively limited scope, they do not provide comparative empirical data for different cases of human–wildlife interactions that confirm their hypothesized structure. Moreover, this and other extant synopses are merely descriptive, rather than elucidating how the individual variables interplay.

An alternative approach to explaining people's similarly polarized and vigorous reactions to different kinds of wildlife, has been proposed by Jürgens and Hackett [2]. Wolves (*Canis lupus*), corvids (*Corvidae*), and spiders (*Araneae*) were chosen as model cases, since they starkly differ in terms of biology, while human relations to all three of these animals exhibit the qualities of being polarized and laden with affect, which are typical of potentially conflictual human–wildlife relations more generally [29–33]. Based on a comparative analysis of scientific as well as cultural literature on these model animals, we showed how encounters with them may confront a person with fundamental questions, i.e., about the role of mankind in nature, and call upon existential motivations, i.e., a want for control over the events in one's life. We hypothesized that the polarized reactions to wildlife can be explained by the different responses to these psychological challenges; while the similar intensity of responses to disparate kinds of wildlife originates from similar mental processes being activated.

This study is designed to assess the adequacy of this conception and to explain in detail the factors and mechanisms that create people's vigorous yet opposing responses to encounters with ecologically disparate wildlife. Our aim is to thus propose a cohesive framework that integrates previous approaches into a comprehensive picture of the human dimensions that act as powerful influences on wildlife populations and their management.

## 2. Methods

We aimed to explore people's interpretation patterns [34,35] with regard to different kinds of wild animals and human–wildlife interactions, with the goal of providing a detailed explanation of their polarized and intense responses. An interpretation pattern is a system of "knowledge, norms, values and interpretations" by which a person understands and reacts to reality [34] (p. 9; first author's translation). As a qualitative research approach is uniquely apt for this purpose [36], we chose to conduct in-depth interviews that allowed us to focus on individual cases and attain a close-up view of the underlying mechanisms [37,38]. We compared participants holding a favorable perspective to those holding

a critical perspective on wolves (*Canis lupus*), corvids (*Corvidae*), and spiders (*Araneae*) as model animals.

By choosing stakeholders of opposing camps and model animals from diverging taxa, we sought to tease out the commonalities and particularities of the different ways of relating to various kinds of wildlife, since "common patterns that emerge from great variation are of particular interest and value in capturing the core experiences and central, shared dimensions of a setting or phenomenon." [39] (p. 235).

### 2.1. Sampling

Twenty human subjects were purposefully sampled based on the maximum variation in the valence of their attitude to the specific model animal as assessed in a short recruiting conversation via phone or email, and on their formal relation to the specific model animal, operationalized by their profession or vocation, e.g., scientist, shepherd, hunter, or environmentalist. Appendix A Table A2 lists the demographic information of the participants. Subjects were recruited via a snowball sampling technique for "locating information-rich key informants or critical cases" [39] (p. 237). Moreover, we located critical cases by tracking down users who had posted expressive commentaries in online news feeds on human–wildlife interactions.

In this way, our sample emerged in the course of the study. The three sub-samples with interviewees on wolves, corvids, and spiders were developed in parallel, i.e., the first, second, third, and all further participants for each sub-sample were recruited at about the same point in the project. Based on the pragmatic criterion for theoretical saturation proposed by Low [40], we discontinued recruiting further participants as a saturation set in with regard to the concepts targeted by the model proposed in Jürgens and Hackett [2] and when in the process of iterative coding [41] (see Section 2.3), themes kept repeating and no genuinely new codes for overarching themes (see Section 3.3) emerged. Theoretical saturation was reached within and across sub-samples. We took this as an indication that heterogeneity of the sub-samples did not lead to significant bias with regard to the research questions. Seven participants were interviewed on wolves, six on corvids, and seven on spiders. German is the primary language of communication for all subjects. 19 subjects were German citizens and one was of Swiss nationality. Participants' original quotes provided in the results section have been translated from German as literally as possible by the first author. Interviewees are identified by a code indicating the focus species in the interview ("W", "C", or "S"), and the order of that participant's interview in the series (e.g., "S5" being the fifth participant having been interviewed on spiders).

### 2.2. Interview Procedure

In-depth interviews lasted between one and three hours and were conducted in person by the first author between June 2016 and October 2020. Subjects were naive with regard to the specific research interests motivating the study and were debriefed afterwards. A question about a meaningful experience [28] the participant might have had with the respective animal kicked off the conversation. The interview then was semi-structured by a set of 15 open-ended questions about participants' personal experience with, general views about, as well as about practical and ideal suggestions for dealing with the model animal and nature in general (see Appendix B). The order and phrasing of questions were adapted to the flow of the conversation and explicit questions were omitted if participants spontaneously brought up the respective themes. Verbal questions were complemented by projective prompts. Participants were also asked to express their thoughts by building configurations of small wooden figures. Projective techniques such as these serve to tap below subjects' conscious filter and to elicit additional insights by addressing striking aspects of those configurations [42,43]. Participants could choose from a total of 80 figures representing wildlife—among them the three model animals, domestic animals, people, cars, fences, trees, buildings, and undefined elements. It was the participants' choice whether or not to use the figures; 15 participants chose to use the figures, and 5 chose not

to use them. The set of questions and projective prompts was the same for interviews on all three model animals.

### 2.3. Analysis

Videotapes of the interviews were transcribed manually by the first author following a scheme developed according to the interests pursued in this study [44]. In addition to explicit statements, the transcripts included all aspects that could be seen as indicating participants' interpretation patterns such as non-verbal communication, e.g., gestures, tone of voice and volume, pauses, as well as elaborate memos with regard to personality traits of the interviewee as apparent by the participants' demeanor before, during, and after the interview [35,45]. The resulting screenplay-like scripts of the interactions [36] were then subjected to thematic analysis [46].

This study is partly of a deductive nature as it empirically investigates the premises made in Jürgens and Hackett [2] as well as tracing concepts established in previous research [17,19,20,26–28]. Simultaneously, it is partly exploratory in regard to discovering potential additional factors and developing a structure of factors impacting human–wildlife relations. Therefore, we employed the qualitative technique of thematic analysis pursuant to Clarke and Braun [46], that "can be used for both inductive (data-driven) and deductive (theory-driven) analyses, and to capture both manifest (explicit) and latent (underlying) meaning" (p. 297), and therefore accords with the two purposes of this study. Analysis pursued a three-step process. We first aimed at an open-ended exploration of themes emerging from the data, "giving a voice to [ . . . ] participants" [41], by extracting their specific views of the model animal and the broader situation of coexisting with this and other kinds of wildlife. These self-assessments—captured in verbatim quotations that were collected to support the respective codes [47]—were complemented with the researcher's transcribed observations and evaluations of the interviews. This first inductive step of analysis yielded a coherent assessment of each participant's cognitive and affectual responses with regard to their relation to the model animal, and to how this relation is embedded within their personal background and lifeways.

In a second inductive step, we coded for themes and patterns that recurred across participants within and across sub-samples. Thus, we derived a list of overarching themes relevant to participants' relations to wolves, corvids, and spiders that are grounded in the data [48]. As a particular part of that analysis across participants and sub-samples, we collected subjects' attributions to each of the three model animals, resulting in a collection of idiosyncratic and recurring appraisals of wolves, corvids, and spiders. We subsequently also coded for attributes recurring across the three kinds of wildlife. Recurring themes and overarching factors across participants and model animals were established via metaphoric correspondences between the key concepts in interviewees' statements as proposed by Benoot, Hannes and Bilsen [49].

In a third step, we purposefully analyzed transcripts for themes that correspond to the concepts pertinent in the literature—e.g., resource conflicts, risks and benefits, membership in stakeholder groups (see Section 1)—and concepts central to previous hypothetical models—e.g., nature-related worldviews and value orientations, motivations of control, and symbolic associations (see above, and [2]). Quotes and observations confirming and challenging these pre-formed codes were collected across participants for each model animal separately and across model animals. Systematically confronting the extant conceptions with the empirical data, accompanied by careful bracketing [50,51], is a means of qualitative hypothesis testing [52,53]. Specifically, for analyzing transcripts with regard to our deductive research objective, we relied on the rationale of Objective Hermeneutics [45], according to which the "latent meaning" (ibid., p. 390) inherent in participants' verbal and nonverbal behavior (captured in the transcripts) serve as a standard against which the adequacy of the hypothesized conceptions can be measured. This deductive third step of thematic content analysis yielded a comprehensive assessment of the meaningfulness of hypothesized conceptions in human relations to wolves, corvids, and spiders compared to

the significance of further overarching themes that emerged through inductive analysis in the second step.

In this three-step process of analysis, we established a typology of overarching factors and idiosyncratic motives that proved impactful in our subjects' relations to wolves, corvids, and spiders and potentially—via a comparison of model animals—for human–wildlife relations in general.

## 3. Results and Discussion

Our comparative investigation of human relations to wolves, corvids, and spiders suggests that for these and perhaps for all kinds of human–wildlife interactions, three layers of contributing factors can be distinguished: a layer of person-specific factors, a layer of species-specific factors, and a layer of overarching factors. A person's response to encounters with a wild animal can be thought of as a composite of a particular instantiation of those factors.

### 3.1. Person-Specific Factors

The layer of person-specific factors is comprised of all aspects that are particular to an individual person's view on a wildlife species (cf. [54]). In this study, we found that our participants' responses to the model species on the individual level are determined by the interplay of two aspects: (1) whether and how a person is affected by the wild animal; and (2) the person's individual and collective identity.

First, the degree, the kind, and the acuteness of being affected by the wildlife species seemed to prompt a person to form a response, but did not determine the quality or intensity of that response (cf. [55]). For example, the negative response of the elderly couple C6 to a flock of rooks flying by their balcony—a seemingly mild way of being affected by corvids—was as strong as respondent C4's reaction who experienced the arguably much more significant situation of witnessing ravens seize a newborn lamb. Conversely, the acuteness of being affected for both C6 and C4 had a huge impact on the intensity of their response. Both participants reported having experienced massive feelings of despair and helplessness during and shortly after the interaction with the corvids, while these states of arousal decreased with a greater temporal distance from the event. Similar effects were reported by participant W6 about his immediate response to versus hindsight of an alleged perpetration of wolves on his horses, and by arachnophobes S1 and S4 with regard to acute versus only recounted contacts with spiders. The direct impact of being acutely affected on the intensity of a person's response corresponds to the fact that a lesser psychological distance to a wild animal led to the enhanced salience of negative facets of its presence, as found by Slagle et al. [56].

Second, with the acuteness, degree, and manner of being affected by a wild animal providing the occasion for a person's response, it seems to be a person's identity that impacts the *quality* of that response to an encounter with wildlife (cf. [57]). We observed two facets of a person's individual identity to be important in determining their response to wildlife: personality traits—as cognizable in the interpersonal situation of the interview, such as irascibility, patience, sarcasm or sympathetic tendencies—and interpretation patterns [34]. People generally refer to interpretation patterns for understanding their reality, including their interactions with wildlife. In this way, people's life themes may be mirrored in their relation to the animal.

For 16 of our participants, it could readily be observed how the themes pivotal to their interpretation patterns defined their relation to the model animals. Moreover, in eight cases, those life themes were observed to give rise to utopian visions of an ideal world in which the biosocial challenges of the present situation would be alleviated and in which a human coexistence with the model wildlife would succeed. For example, participant C4 is a shepherd keeping a small flock of ancient races of sheep. Her vocation is educating children on the biodiversity in traditionally managed grasslands, counteracting the "profit-oriented" mainstream mindset which, in her view, alienates humans from their roots in nature. She is

deeply frustrated by the many administrative hurdles and economic challenges in addition to the natural imponderables which she meets in her work. Her accumulated frustration found a tangible target when she experienced unexpected depredation of ravens on a hand-raised lamb. Her previously positive attitude towards corvids reversed completely. She felt the ravens were akin to "rapists" violating her fosterlings and treating her "good cause" in a disdainful manner. In a rush of rage, she wrote to authorities and environmental organizations, demanding the culling of ravens since "when corvids or wolves misbehave, they must be disciplined." When the first author interviewed her half a year after the event, she conceded that she had somewhat overreacted in the heat of the moment, and to have scapegoated the ravens for the hardships which she had faced all along.

It was obvious how C4's feeling of being vulnerable and helpless in regard to following her mission in the face of seemingly overpowering adversity was reflected in the terror of the ewes and lambs being exposed to the ravens' attack. Accordingly, the utopia she aspires to is one in which an intimate communion of humans and nature will have come to fruition. She envisions that her flock and other livestock may roam freely in a fenceless landscape—symbolizing her own liberation from legal and administrative restrictions.

Table 1 explains for the 15 other participants how their views of the respective model animal are embedded within their life themes, and how these life themes, in turn, may inspire visions of a utopian future of human–wildlife coexistence. As commonalities are evident between subjects, we identified three ways in which life themes shape participants' relation to the model animal and inform their utopian visions. These commonalities and the systematic mappings between life themes and visions are discussed in Section 3.3.3, as they indicate how participants' individual perspectives are interlocked with overarching trans-personal patterns. For respondents W5, C1, C6, and S2, a relation between their life themes and their view of the model animals could have been constructed through interpretation, but these participants did not offer such a correspondence in their own words. Therefore, we decided not to list them. All of these respondents except C6 felt personally unaffected by the model animal and reported not to hold a strong opinion regarding the respective species. This accords with the assumption that affectedness provides the basis for taking a stance towards a wildlife species.

**Table 1.** Participants' life themes within which their relations to the respective model animals, and their utopian visions that include ideals for managing wildlife, are embedded. "~" denotes that for a given subject a clear inclination for a particular theme or vision can be inferred from the context and tone of the interview yet was not explicitly declared.

| | Utopian Visions | | |
| **Participants' Life Themes through the Lens of Which They View the Relation to the Wildlife Species** | *An Extended Family Embracing Humankind, Wildlife, and Nature* | *A Traditional Human–Nature Relationship Based on Sustainable Use* | *A Cohesive Society Based on Tribal Structures, Solidarity, and Respect* |
|---|---|---|---|
| *The relation of humans to the wildlife species pinpoints the relationship between humans and nature more generally.* | | | |
| W1 | His life theme, as an influential animal rights lobbyist, is fighting against a presumed annihilating intention that people direct towards animals in general which gets actualized specifically in anti-wolf sentiments. | The Kantian imperative acts as the ethical base for treating beings of all species as if they were family members. | |
| C2 | He charges modern people's allegedly ubiquitous unwillingness to accommodate inconveniences of all kinds, which also causes them to oppose the presence of corvids in their proximity. | | The revival of traditional farming be a cure for environmental, social and health issues. |

**Table 1.** *Cont.*

| Participants' Life Themes through the Lens of Which They View the Relation to the Wildlife Species | Utopian Visions | | |
|---|---|---|---|
| | *An Extended Family Embracing Humankind, Wildlife, and Nature* | *A Traditional Human–Nature Relationship Based on Sustainable Use* | *A Cohesive Society Based on Tribal Structures, Solidarity, and Respect* |
| C5 — She reports to have suffered immensely from human hard-heartedness in her life and sees her suffering as being mirrored in the crows who are treated cruelly by people. | A Christian paradisiac ideal of harmony among all beings. | | ~ |
| S1 — ~ | All species be released from the metabolic cycle, so a cruelty-free interbeing may ensue. | | |
| S4 — Being a Christian deacon, her life is oriented towards worshiping the whole of creation. In her view, her arachnophobia and the controlling measures she takes starkly contradict her moral code. This quandary distresses her on a daily basis. | | ~ | |
| S6 — Perfused by epistemological striving, he considers fractality a building block of the universe. Perceiving it in a spider's web struck him in an epiphany about the oneness of all beings, thus soothing his aching search for humanity's place within nature. | ~ | | |
| S7 — As a scientist investigating spiders, he still regards them as only one link in the global ecosystem. However, he views the human–spider relationship as the epitome of human intrusiveness that needs to be obverted. | | ~ | |
| *The wildlife species evidences the necessity for humans to manage nature, particularly to control wildlife population density.* | | | |
| W2 — He claims that as a general rule of existence, organisms and technical developments will propagate boundlessly and must be regulated by humans—including wolves, who need to be controlled though hunting. | | | |
| W6 — He shows the demeanor of a traditional patriarch towards his wife and horses whom he regards as extended family. He owns a large property whose ecological value he carefully maintains according to ecocentric values from which wolves and magpies are selectively exempted. He advocates for the eradication of all wolves from Germany and culling of magpies. | | ~ | |

**Table 1.** *Cont.*

| Participants' Life Themes through the Lens of Which They View the Relation to the Wildlife Species | Utopian Visions | | |
|---|---|---|---|
| | *An Extended Family Embracing Humankind, Wildlife, and Nature* | *A Traditional Human–Nature Relationship Based on Sustainable Use* | *A Cohesive Society Based on Tribal Structures, Solidarity, and Respect* |
| C3 — Being a passionate hunter and hunting lobbyist, he is devout to the conviction that in the present landscape which has been significantly altered by humans, management though hunting is pivotal for establishing and maintaining a balance and diversity of species. He takes corvids, as hemerophiles, to be the epitome of this general principle. | | Restoring a diversity of game that parallels those of traditional mid-European hunting grounds from "my granddad's times" | |
| *The ways in which humans deal with the wildlife species allegorize general deficits in society and in the political arena.* | | | |
| W3 — An independent-minded entrepreneur, he scornfully accuses politicians and authorities of incompetence and hypocrisy. The ways in which the responsible parties deal with wolves evidence the general issues of wasted taxpayers' money, crookedness among his fellow hunters, and societal egocentrism. | | ~ | |
| W4 — Portraying himself as a responsible hunter, he loathes the legal framework restraining hunters from enacting their free will within their hunting ground. Wolves epitomize the encroachments on hunters' sovereignty. | | | |
| W7 — Based on his experience in the political arena as a member of parliament and wise-use lobbyist, he proposes that debates on wolves evidence fundamental problems in the German mindset and political system: an interest in power instead of in resolving practical challenges; putting ideology first and reality-checks second; an alienation of the societal majority from urban lifeways; and a failure to act on humans' quasi-sacred responsibility for managing and thus maintaining the cultural landscape. | | | |
| C4 — An ecologically minded shepherd providing environmental education for children, she meets many administrative hurdles and policy-induced economic challenges in addition to the natural imponderables such as predation by wolves and corvids on her sheep. She suffers from the system being profit-oriented rather than supportive of ecologically and socially meaningful vocations such as hers. | ~ | Human society being intimately connected with nature in a cultural landscape allowing unrestricted roaming for livestock and wildlife. | |

**Table 1.** *Cont.*

| | Utopian Visions | | |
| Participants' Life Themes through the Lens of Which They View the Relation to the Wildlife Species | *An Extended Family Embracing Humankind, Wildlife, and Nature* | *A Traditional Human–Nature Relationship Based on Sustainable Use* | *A Cohesive Society Based on Tribal Structures, Solidarity, and Respect* |
|---|---|---|---|
| S3 — He sees an allowing and respectful handling of spiders as an allegory for the appreciation and loyalty in human society that is to be aspired, but allegedly has deteriorated in modern times. | | | He extrapolates his childhood experience of growing up on his grandparents' farm to a mythical past and to an envisioned future of humankind that are characterized by granting autonomy yet ensuring solidarity to all individual members of the social group. |
| S5 — Having grown up in North Rhine-Westphalia and worshiping the Viking culture, he holds strong views with regard to what qualifies a worthy person. He purports that a person's nature is evidenced through the way in which they treat spiders, which also parallel the ways in which they will treat other animals and their fellow humans. | | ~ | A vision for society based on assumptions about the character of the Westphalian "race" and Viking tribalism: loyalty, toughness, self-assertion, and respectful demeanor to all life forms. |

For some of our participants, we observed that their interpretation patterns and thus their response to an encounter with the model species were also partly shaped by their collective identity, e.g., the pertinent socio-demographic variables (listed for our participants in Appendix A Table A2) and their subscribing to a particular stakeholder group's view of the model animal (cf. [55]). For example, hunters and farmers have been shown to entertain a specific perspective on nature in general and wildlife in particular [58–61]. Moreover, different sociodemographic groups such as the elderly, less well educated, rural, or right-wing populations tend to show more negative views, particularly of problem wildlife, than younger, educated, urban, or leftist people [13,15,62–64]. However, due to the small sample size, systematic relations between sociodemographic variables and participants' attitudes could not be established. Moreover, they were not the focus of this study. However, we may state that in our qualitative procedure we observed (similar to Grima et al. [65]) that the influence of subjects' individual perspectives dominated over pertinent group-related interpretation schemes. For example, shepherd C4 exhibited an overall rather critical attitude towards human coexistence with corvids and wolves. Some of her arguments, e.g., calls for effective measures for controlling wildlife, parallel the demands on the part of livestock farmers and their political representatives [66,67]. Yet, her stance is far from being a blunt recitation of group norms. Rather, her view is ambivalent and profoundly nuanced in neat accordance with her multifaceted individual motives, namely her ecological ethics. As such, our results suggest the influence of collective identity works through and is mediated by a person's individual identity.

In summary, the person-specific layer of factors constitutes a person's disposition to react in a certain way to encounters with wildlife. Specifically, individual and collective identity, as manifested in a person's biography, influence the valence of the response, thereby providing the ground for explaining the polarization of responses. Therefore, the layer of person-specific factors mediates the impact of the factors of the further layers on a person's ultimate reaction to wildlife.

### 3.2. Species-Specific Factors

Species-specific factors constitute the aspects that wild animals contribute to human–wildlife relations. Animals take an active role in co-creating human–animal interactions [5,9,11,68,69]. However, the behavior of wild animals informs human responses in an indirect manner, namely through the gateway of people's mental images of the animal. Mental images are constituted by stereotypic representations of the animals as perceived and evaluated by humans [2]. These representations are not comprised of unadulterated biological facts but are assumptions and interpretations of the animals' ecologies, people's definitions of challenges and benefits in coexisting with the animal, attributions, and symbolic associations [2,4,56,70,71].

In Section 3.1, we considered the finding that people's biographies provide the context within which responses to encounters with wild animals are embedded (see Table 1). Analogously, we found that participants' mental images of the model animal concorded with their general interpretation patterns (see also [72]). As interpretation patterns are person-specific, so are the representations of wildlife. Consequentially, when seen through a person-specific lens, the same ecological facts gave rise to opposing representations of wildlife. For example, respondents W2, W5, and C3 held the general view that hunting by humans is pivotal for establishing and maintaining a balance in ecosystems. Correspondingly, the assumption that wolves and crows would naturally overpopulate if not lethally managed, was part of their mental image of wolves and crows. Conversely, W3 and C2 presume that wildlife populations self-regulate; therefore, the participants' representation of wolves or corvids included the idea that their numbers would naturally find a point of saturation.

In Table 2, we listed the detailed attributes that comprise the participants' mental images of wolves, corvids, and spiders. For a straightforward overview, we juxtaposed rather critical and rather favorable attributes as elements in representations. However, only two participants held representations of solely critical or solely favorable elements: W6 mentioned only negative attributes of wolves, while W1 named solely positive qualities. The majority of 18 participants exhibited more nuanced representations, e.g., consisting of mostly critical, but some favorable attributes or vice versa.

**Table 2.** The elements constituting participants' mental images of wolves, corvids, and spiders.

| Themes | Wolves Critical | Wolves Favorable | Corvids Critical | Corvids Favorable | Spiders Critical | Spiders Favorable |
|---|---|---|---|---|---|---|
| useful–not useful | Wolves cause damage, while their presence serves no purpose. (W2, W4, W5, W6, C2, C4) Being apex predators both, wolves and human hunters compete for game and potentially render hunting unnecessary and unfeasible. (W2, W3, W4, W5, C3) Wolves are not good managers of game populations. (W2, W6) | The presence of wolves may be beneficial for hunters as it enriches the ecosystem and renders the experience of the hunt more exciting. (W3, W7, C2, C3) | Corvids are vermin causing nuisances. (C3, W4) Corvids cause noise. (C1, C2, C5, C6, S4) Corvids empty trash cans. (C1, C6) Corvids befoul pavements and cars. (C2, C6) Corvid depredation endangers small game and other bird species. (C3, C6) | Rooks are potentially useful, e.g., when devouring crop pests. (C3) | Spiders are considered vermin and a potential "problem of hygiene"(S6) to get rid of as part of tending to one's home. (S4, S5, S6) Spiders cannot be put to use by humans. (S4) | Spiders are useful and diligent, e.g., in devouring pests such as mosquitoes. (S1, S4, S5, S6) Spiders are quiet beings. (S4) |

**Table 2.** *Cont.*

| Themes | Wolves Critical | Wolves Favorable | Corvids Critical | Corvids Favorable | Spiders Critical | Spiders Favorable |
|---|---|---|---|---|---|---|
| | Wolves are a constant threat to livestock despite protection measures. Their presence impedes farmers from responsibly caring for their livestock. Given the emotional connection of farmers to livestock, they threaten farmers' mental wellbeing. (W4, W6, W7, C4) | Wolves' depredation on livestock is to be accepted as a natural phenomenon. (W1, W3, C4, S2) | Corvids kill newborn livestock. (C4) Corvids devour seedlings. (W4) | | | |
| dangerous–harmless | Wolves are dangerous to humans. (W2, W5, W6, C2, C3) | Wolves will only be dangerous to humans under exceptional circumstances, e.g., if injured. (W1, W3, W4, W7, S2) | Corvids could harm humans, e.g., with their strong beak. (C1, C4) | | Spiders evoke fear due to their (seeming) ability to harm humans. (S1, S3, S4, S6) | Spiders are harmless. (S2, S4, S6, S7) |
| | Wolves emanate a sense of constant, omnipresent threat. (W6, C4) | | The presence of corvids emanates a sense of threat. (C4, C6) Corvids' presence is reminiscent of Hitchcock's *The Birds*. (C4, C6) | | Spiders can appear anywhere any time; they seem to be omnipresent—thus scaring arachnophobes even if not seen. (S1, S4, S6) | |
| | Wolves reproduce boundlessly if not controlled. (W2, W4, W5, W6) Wolf numbers must be capped. (W3, W7) | | Corvid numbers have risen constantly and significantly. (C3, C4, C6) Corvid numbers must be reduced to keep a natural balance. (C3) | | | |
| (un)controllable | Wolves' behavior cannot be controlled. (W2) Wolves are impudent, must be "kept in check". (W2, W6, C4) Wolves do not flee in human presence (W7) | Wolves cannot be domesticated. (W1) Wolves are nocturnal. (W2, W3) Wolves are shy and evade human presence. (W1, W2, W4) | By flying, corvids master the third dimension, making them even harder to control. (C4, C6) Corvids behave impudently in coming close to humans. (C4, C6) | | The unpredictability and speed of spiders' movement are unsettling; particularly their sudden appearance near to a person is fearsome. (S1, S2, S4, S6, S7) Killing spiders is an involuntary response for restoring control. (S1, S4, S6) | Spiders' mastery the third dimension with their web-weaving is similar to a superpower. (S4, S6) |
| | | | | Corvids' agency is salient since they appear to be always on the go, playful, and full of joie de vivre. (C1, C5) | | Spiders exhibit deliberate and intentional behaviors. (S3, S6) |

**Table 2.** *Cont.*

| Themes | Wolves Critical | Wolves Favorable | Corvids Critical | Corvids Favorable | Spiders Critical | Spiders Favorable |
|---|---|---|---|---|---|---|
| (un) social | Wolves' living in social groups makes their impact particularly problematic. (W2, W4) | Wolves are caring, social beings and have families, just as humans do. (W1) | Corvids wrangle with each other. (C6) | Corvids are social beings exhibiting loyalty, loving relationality, and caring towards their kin and other species, including humans, Thus, they are models for humankind. (C3, C5, S6) | Spiders are utterly alien to humans in their ways of being; no mutual understanding or communication is feasible. (S2, S4, S5, S6, S7, C1) | Humans and spiders may share a sense of mutual apperception that, at least on the part of the human, can be seen as relationality. (S3, S4, S5, S6) |
| (un)aesthetic | Wolves are unaesthetic. (W4) | Wolves are beautiful. (W2, W7) | Corvids' blackness is a salient and potentially uncanny feature. (C1, C3, C4, C6, S4) | Corvids are beautiful, impressive beings, particularly because of their size. (C1, C5, C6) | Spiders are not seen as being cute by most people. (S2) Spiders are prototypically represented as being dark. (S4, S6) | Spiders are aesthetic beings. (S2, S7) |
| ambivalent fascination | | Wolves are fascinating, numinous, awe-inspiring beings. (W1, W7, C2, C3) | | Corvids are fascinating to watch. (C1, C3, C6) Corvids are numinous, awe-inspiring creatures. (C5, C6) | Spiders evoke a distancing response (a mild sense of disgust and fear) even in people not particularly opposed to them. (S2, S6, S7) | Spiders and their lifeways (e.g., web-weaving) are fascinating, numinous, awe-inspiring, and daunting. (S1, S2, S3, S4, S5 S6, S7) Spiders' strangeness bestows a sense of specialness onto humans associated with them. (S2, S3, S4) |
| intelligent and capable | Wolves are capable of calculating, strategic moves. (W6, W7) | Wolves are intelligent and can learn quickly. (W1, C4) | | Corvids are intelligent, knowledgeable and wise. (C1, C2, C3, C4, C5, C6, W4, S6) Corvids appear to have a perspective of their own. (C1, C3, C5') Corvids have epistemological interests. (C1) | | Spiders may be endowed with an ancient wisdom. (S1, S6) Spiders exhibit a great deal of creativity and deliberate artistry in web-weaving. (S4, S6) |
| | | Wolves are opportunists and adapt to different circumstances. (W3, W7) | | Corvids are opportunistic profiteers of the human-made landscape and bohemians exhibiting a "toughness" in getting along. (C3) | | Spiders are persevering in the face of adversity. (S4) |

**Table 2.** *Cont.*

| Themes | Wolves Critical | Wolves Favorable | Corvids Critical | Corvids Favorable | Spiders Critical | Spiders Favorable |
|---|---|---|---|---|---|---|
| morally condemnable | Wolves perform excessive surplus kills. (W4, W6) Wolves kill particularly cold-heartedly, cruelly, as "killers" (W2; W4, W6, W7) Wolves are akin to "criminals". (W6, C4) Wolves are ever-hungry beasts. (W2) | | Corvids are cold-blooded "killers" of lambs and small game. (C3, C4) Corvids are similar to "terrorists" and "rapists". (C4) | | Spiders evoke an amorphous impression of being evil creatures. (S1, S5, S7) Spiders pursue a predatory lifestyle. (S4) | |
| disgusting | | | Corvids are associated with filth, e.g., waste dumps and decaying corpses. (C1, C3) | | Spiders evoke disgust, particularly due to the shape and proportions of their bodies. (S1, S3, S4, S6, S7) The larger the spider, the greater feelings of disgust and fear. (S1, S2, S4, S6, S7) | |
| (not) belonging | Wolves do not belong to and should be kept out of Central Europe. (W6) Wolves have their place in nature, not in the cultural landscape. (W2, W3, W4, W7, C2) | Wolves are an integral part of the ecosystem and have been and are meant to be part of Central European landscapes. (W1) | | As hemerophilic wild animals, corvids populate an intermediate realm between nature and the human sphere. (C1, C3, C5) Rooks' presence is an indicator of a healthy ecosystem. (C2) | | |
| nature | Wolves epitomize the fact that nature can be cruel. (C4, C5) | Wolves are symbols for pristine nature, wilderness, and for the resilience of nature. (W1, W7) | Corvids can be brutal and thus evidence the fact that nature can be cruel. (C2, C3, C4, C5) | | | Spiders are primordial beings, and symbols for life, i.a., due to the evolutionary persistence of their class. (S3, S5, S6, S7) |
| poise | | Wolves are symbols of strength and assertiveness. (W1, W7) | | Corvids appear regal and self-conscious. (C1, C5) | | Spiders have a lordly appearance. (S2, S2) |

Simultaneously, the juxtaposition of attributes presented in Table 2 also evidences that participants' mental images exhibited systematic similarities. Not only did participants' views concur within the rather critical and rather favorable perspectives but certain features emerged in both critical and favorable participants' mental images. These features were just expressed in opposing, person-specific ways, depending on a person's interpretation pattern. For example, the issue of potential harmfulness of wolves, corvids, and spiders was raised by virtually all participants. However, the idea that wolves, corvids, and spiders may harm humans was affirmed by critical participants and rejected by favorable participants.

Moreover, systematic similarities also occurred across the three model animals. We discuss these findings in depth in Section 3.2.4. First, we present synopses of the mental

images of wolves, corvids, and spiders. Where available, we reference studies in which corresponding concepts are discussed.

### 3.2.1. Wolves

If one takes a critical view, wolves are useless harmful beings threatening humans' economic and potentially physical wellbeing, who do not "fit into the cultural landscape" and have been "eradicated for a reason" as W4 suggests (cf. [6,73]). Since wolves live secretively yet are said to have lost their fear of humans and may strike unexpectedly, they emanate a sense of omnipresent threat (cf. [74]). Because their behavior evades human control and their alleged tendency to reproduce boundlessly in a way supposedly "characteristic of their species" (W5), wolf-critical participants argue that wolf numbers must be controlled through hunting (cf. [75]). Rising wolf densities and constant depredation on livestock for some participants make wolves "embodied consequences" (W7) of humans' failure to neatly manage the landscape. "Keeping tabs on wolves" (W2) seems even more important to critical participants since they see wolves' behavior tainted with "implicit moral judgment" [66] (p. 286). Wolves' predatory behavior is said to be particularly cruel and "senseless" (W4); they seem to exhibit a "lust for killing" either independent from actual need (W6) or attempting to quench their "insatiable" hunger (W2). Wolves are described as "criminals" (W6) and likened to "rapists" (C4), violating innocent lambs (cf. [4,76]). Perceptions of wolves as insatiable, cold-blooded killers and as a symbol of destructive masculinity are pertinent in the wolf-critical parts of contemporary and past Western societies [2,57,66,76–80].

Conversely, wolf-favorable participants viewed wolves as intelligent and "caring" (W1) social beings who are granted a "right to be there" based on "environmental ethics" (W7) that bestow existence rights to wildlife having traditionally belonged to Central European ecosystems (cf. [73,81,82]). Moreover, many wolf-favoring participants venerate wolves as epitomes and wardens of wilderness, i.e., nature untouched and unbridled by humankind. Thus, the resurgence of wolf populations is viewed as an encouraging indication of the resilience of nature restoring its balance and defying human encroachment and desecration. Accordingly, challenges in human–wolf coexistence are viewed as "ecological necessities" (W1) to be tolerated whilst wolves inadvertently pursue their life ways [70]. In this reading, wolves' uncontrollability is cherished and wolves are viewed as "alpha animals" (W1) who demonstrate the limits of humankind's power.

### 3.2.2. Corvids

Corvids were perceived as being "uncanny" (C6) and "scary" (S4) by many corvid-critical participants. The attribution of a "menacing" (C1) intent to corvids (cf. [70]) may originate from their impressive size and, particularly, their dark exterior (cf. [83,84]). "That blackness, that is interfering, yes, frightening or, I'd say, fearsome." (C6). References to Hitchcock's movie *The Birds* were pertinent (cf. [2,85]), a fortiori due to many corvid species' appearing in large flocks and—as many participants claimed—growing numbers. In a similar vein, crows' and ravens' depredation on small game and, allegedly, on young livestock is described as particularly "unsettling" (C3) for the human observer. Similar to wolves, corvids are considered "vermin" (W4) from an utilitarian perspective (cf. [86]) and when judged in moral terms, they appear as "hostile" "murderers" (C4) when feeding on lambs, and as being "terrorists" (C4) and "impudent" (C6) when fearlessly approaching humans.

In contrast, from a corvid-favorable view, corvids' seemingly "brutal" forms of foraging and the annoyances they cause "belong to nature" (C6), and the potentially scary facet of the birds instead appear as awe-inspiring (cf. [87]). Corvid-favorable participants adored corvids' numinous, i.e., "mystic", "majestic" (C5), and seemingly "self-conscious" (C1) demeanor and rejoice in them "always being on the go" (C1). Their witty looks ostensibly make them agents of their own life (cf. [88]). Corvid-favorable participants "respect" (C3) and find "heartrending" (C5) their evasion of human control (cf. [2,89]). Yet, due to their prudence and proverbial intelligence that is integral to all participants' mental image of

corvids (cf. [85]), they were seen as being able to become partners in relationality with humans [70]. Furthermore, due to their apparently "loving" care (C5) towards their kin, they were deemed models from whom "humans could learn a great deal" (S6).

### 3.2.3. Spiders

The most salient aspect that made participants hold critical views of spiders is their utterly alien nature to humans with respect to their mind and physique (cf. [90]). Their "unconceivable" shape, e.g., their possession of "too many legs" (S4), automatically evokes fear and disgust, even in some non-phobic participants. Moreover, spiders' "unnatural" (S4, S7) way of being in comparison to humans and other animals to whom humans may intuitively relate, disables any kind of communication. The absence of any discernible display of emotions or other signs of a mental life on the part of spiders (cf. [91,92]) in conjunction with their obvious intentional behavior that "requires a reaction" (S4) by humans is "disturbing" (S7), not only to arachnophobes. Participants described the speed and uncontrollability of spiders' movements as challenging to deal with (cf. [93]). They felt a loss of control in the face of spiders' ability to appear "all of a sudden" (S4) in a person's proximity unreckoned (cf. [94]). Spiders thus cause a sense of acute "alarm" or latent "stress in one's core" (S6), "since worse than a spider you see is the spider you don't, as she could reappear any time. [ . . . ] Not-knowing is so much worse than knowing." (S4). Killing a spider is oftentimes a quasi-intuitive mechanistic reaction in the heat of the moment for seeking relief of the fear and for restoring control. It also sets the scala naturae right, since the spider as an allegedly simple being no longer "holds power" (S4) over the human (cf. [70]).

Spider-favorable subjects, to the contrary, hold a thorough fascination for spiders who in their inapproachable strangeness may still be regarded as "aesthetically appealing" (S7) and inspire awe due to their "lordly" (S2) demeanor. They are not only praised as "useful animals" because they eliminate insect pests. Given that they "don't even have an actual brain" (S6), it is even more impressive that they exhibit complex goal-directed behaviors. For example, their web-weaving was considered by some participants as "creative artistry" (S6) that equals and surpasses human skills (cf. [95]), while their mastery of the third dimension inaccessible to humans is akin to a "super-power" (S4). Spiders' deliberate actions to some even seemed to be informed by a "higher plan" (S6). As the evolutionary emergence of spiders as a biological order dates far back, they can be regarded as a "primordial" form of being (S7), even as an epitome of "life itself" (S3) or a medium of "life energy" (S6). Against the backdrop of spiders' supposed "deep wisdom" (S6), and despite their inability to communicate in a human-like form, there may still be ways in which humans and spiders can relate, e.g., on a "spiritual" (S3) level [70]. In the wake of such an infinitesimal relationality, humans who associate with spiders may enhance their status by transferring onto themselves the same traits that evoke critical responses in other people. The "extremeness" of spiders identifies their owners and admirers as being endowed with "hardcore" (S3) personalities.

### 3.2.4. An Overlap of Species-Specific Mental Images

The results of this study expand extant proposals for species-specific factors which determine the likelihood of an animal to be deemed a "conflict species" [96] (p. 159) and which define the characteristic tone of a person's relation to the animal. Kansky and Knight [19] have proposed "observed abundance" as a "perceived species characteristic". The multifaceted elements found to comprise people's mental images in our study illustrate that perceptions of wildlife are much richer than just observed abundance. For example, participants wove facets into their mental image of the three species—e.g., spiders' quiet form of existence, or corvids' ostensible shrewdness—that constitute delicate observations of the animals' particular ecology. In this respect, our results concur with Kellert [97] and Serpell [98], who identified an elaborate list of fundamental and concrete qualities of wild animals that impact people's attitudes to them. Our findings go beyond these collections of

attributes by incorporating opposing stances to any one species, and by showing how a person's mental representation is the medium through which the species-specific factors take effect. In line with Lescureux and Linnell [5] and Jürgens and Hackett [2,4], we argue that certain aspects of animals' behavior, e.g., their manner and speed of movement or ways of foraging, automatically cause the individual to heed to them in the process of forming a representation of these animals. Thus, animal ecology may serve as a point of reference, while people's interpretation patterns stemming from their individual and collective identities (Section 3.1) determine the shape of the representation that is built from a selective and idiosyncratic appraisal of the ecological facts.

Due to this systematic interaction of a person's mental processes with wildlife ecology, certain themes may figure consistently in people's mental images of an animal and thus be species-specific, while simultaneously take different forms for different people and thus be person-specific.

The pivotal role that representations play in people's responses to wildlife is also evident in the fact that mental images overlap between ecological disparate animals. Our results show that people's mental images of wolves, corvids, and spiders were composed of largely the same themes. In Section 3.3, we explain that these concurring perceptions of the model animals entwine with the overarching themes of human–wildlife relations. Exactly how these themes were instantiated was species-specific, yet the human mind seems to systematically extract particular aspects from the richness of wildlife ecology, thus portraying the "animals-as-constructed" [99] as being more similar than their corporeal counterparts. For example, all three animals were described as awe-inspiring and somewhat numinous. Albeit this impression flows from different sources for each of the three animals, all of them attract appraisals of "vastness, need for accommodation, [ . . . ] threat, beauty, exceptional ability, virtue, and the supernatural" that have been found to elicit awe [87] (p. 297). For wolves, a sense of awe seems to be caused by their charisma as large predators that may make "the entire forest fall silent as they approach" (C2). For corvids, awe is elicited by their large size in comparison to other endemic birds, by their blackness and "majestic demeanor" (C5). For spiders, their alien physique and ways of living, as well as their "lordly appearance" (S2), are described as inspiring awe, including as sense of "respect" and reserve (S7). Therefore, we find that it is not wildlife per se, but wild animals seen through the lens of human representations, who are the contributors of species-specific factors impacting human–wildlife relations [73,80].

*3.3. Overarching Factors*

Across time and cultures, wolves, corvids, and spiders have been represented in astoundingly similar ways in ancient mythology and recent cultural renderings despite striking ecological differences between these animals [2]. In the previous section, we showed that such a congruence also exists in participants' active representations of wolves, corvids, and spiders, i.e., the mental images they employ for making sense of encounters with these animals. We propose that this is due to the fact that wolves, corvids, and spiders raise the same fundamental questions about the human condition. This proposal is supported, first, by respondents' readiness to draw comparisons between different cases of human–wildlife relations. Even though each interview focused on only one of the model animals and the interviewer only disclosed the comparative intent of the study afterwards, some participants spontaneously mentioned other wildlife species, indicating that their conclusions drawn with regard to the model animal pertained to human–wildlife relations more generally. Cross-references to wolves were particularly common, probably due to their presence in the public media. For example, C1 proposed that modern humans exhibit an unwillingness to accommodate to crows' needs and concluded "It's the same we see with wolves!".

The idea that different kinds of wildlife raise the same fundamental issues is further corroborated by the three life themes and utopian visions in which participants' views of the model animals are embedded (Table 1): (1) participants' relation to the model animal is

a reflection of how they relate to the natural world more generally, (2) participants view the model animal as a paradigmatic example for the alleged fact that humans must regulate nature and wildlife population density, and (3) participants view the way in which the model animal is treated as pointing out what is wrong with human society in general, e.g., a general "lack of considerateness" (C5), or "effeminacy" (S5). In all of these cases, subjects took the human relation to the model animal as being similarly emblematic of dynamics in much larger contexts. Those life themes fed into corresponding visions that paint a utopian picture of how the current challenges may be alleviated. These visions also emerged in similar ways in interviews on the different model animals.

In the following sections, we provide details of two fundamental questions that wolves, corvids, and spiders—and arguably by many more kinds of wildlife—raise and explain how this occurs.

### 3.3.1. The Question of Humans' Place in Nature

An integral facet of participants' mental images of all model animals is their status of being symbols for nature. Wolves in particular have pertinently been described as epitomes of wilderness [6,66,79,90,97,100,101]. Our results show that the same is true for corvids and spiders, suggesting that potentially any kind of wildlife may gain that emblematic power of representing the whole of nature when they become salient to humans in positively or negatively meaningful encounters [28,58]. In particular, this implies that through interactions with wildlife, humans may attempt to configure their relation to nature. At some point in all interviews we conducted, participants spontaneously brought up a fundamental question regarding the place of humans in—or above—the natural world. The specific issues they raised concerned (i) the legitimacy of human use of natural resources, and the prioritization of human claims over other beings' needs in situations of resource conflict; (ii) the degree to which humans can be said to be part of and connected to nature or distinct from it; (iii) whether humans are endowed with a particular responsibility to care for or manage nature; and (iv) how non-human beings are to be viewed and treated, e.g., as individuals and persons, or as populations and species.

Some of these themes relate to well-known concepts in the environmental psychology and ethics literature, such as the anthropocentrism–ecocentrism–biocentrism divide [102–104], mutualistic versus dominion-oriented stances [105,106], or atomistic vs. holistic perspectives on non-human beings [107,108]. Others correspond to models employed in wildlife management, e.g., the separation versus coexistence model [1,6,109]. Analogous to the fiercely fought academic battles over these perspectives, participants also disagreed in their stances towards those issues, dependent upon their individual backgrounds (Section 3.1).

*(I) Hegemony* vs. *considerateness:* Anthropocentric use-oriented perspectives are pitted against eco- and biocentric stances in our sample, as in society [110–112]. For example, arachnophobe S4 relates her dislike of spiders directly to their alleged lack of usefulness: "If I were God, I'd create spiders less disgusting and [ . . . ] as having some great benefit for humans. Cows don't disgust us, because they give milk. Bunnies don't disgust us, since we can pet them. Dogs don't disgust us [ . . . ], because they guard us. Spiders can eat insects [ . . . ], alright, but [ . . . ] spiders would need to be capable of something cool to benefit us." In striking opposition to this view that isbased upon utility to human beings, spider-enthusiast S7 denounces people's treatment of spiders through anti-anthropocentric sentiments: "Humans are like a tumor. They expand, seize animals' habitat, and when animals out of necessity use allegedly 'human' spaces, they are upset and go so far as to repel or exterminate them."

*(II) Distinctiveness* vs. *connectedness:* Similarly polarized stances exist with regard to humans' being distinct from or connected to nature: C5 puts her love of crows and all other animals in a wider context of a quasi-paradisiac divine intention: "God envisaged unity to exist on earth, and intended for us to respect our animal neighbors as we respect ourselves. He wanted us to live in a communion with nature!" That is why she objects to any way in which humans claim supremacy over animals, e.g., in using them for food: "To me, the idea

of deciding the fate of an animal is horrifying. I am not entitled to do that! [ . . . ] That is a blatant encroachment!" However, the perspective of hunter C3 shows that viewing humans as part of nature is not necessarily in contradiction to assuming a prescinded status: "I do believe that we, as humans, should not view ourselves as separate; we are part of nature. We constitute an evolutive factor for game species—and we have been given a mandate here! For example, the crows we don't get to shoot, they are fitter, smarter, than the ones we catch. [ . . . ] And since we are part of nature, I also have a right to take an animal that I want to use. [ . . . ] We clearly are at the apex position of the food chain, we have been endowed with our intelligence for a reason! Yet, therefore, we also carry responsibility."

*(III) Managers* vs. *caretakers:* A responsibility originating from the unique status of humans is stressed by many participants, both from the connectedness and the distinctiveness camp. Yet, they disagree on whether that responsibility ought to be "caringly" used to limit human influence on natural processes "for the good of the animals" (W1), to "regulatively" (W6) ensure their unimpeded flow, or to master and manage them. Many hunters claim that human interventions are needed to "maintain balance" (W4) in an ecosystem. Conversely, W3, a hunter himself, claims that "wolves, lynx, all animal species, have had a right to exist far before humans became civilized, since things just align and work well in nature. Yet, when humans try to regulate nature, [ . . . ] things go downhill." He explains that human attempts to manage nature better than predators are doomed to fail, since people are motivated by their own interests and not by what is best for the ecosystem.

*(IV) Populations* vs. *Individuals:* Finally, participants differ in regard to whether their focus is on animals as "individual beings" (W1) or on animals as species or populations with the seat of animals' essence being an abstract entity whose instantiations are the individual organisms. The latter stance is taken by many hunters in our sample who characterize the chase as "a competition about 'who is better? Me or the game animal? Who is smarter?'" (C3). Logically, the competitor cannot be the individual animal, who—when "outsmarted" (W7) by the human hunter—is annihilated; instead, the competitor must be some kind of master spirit (cf. [113]) who counts the score for the animal party. Likewise, when participants demand that humans need to "keep tabs on wolves" (W2) or "ravens need to be taught how to behave" (C4), they refer to an idea of learning on the species level where casualties on the level of individual animals mark the progress of the training. Conversely, other participants view animals as unique creatures, each endowed with inherent worth that flows from their personhood. S5, for example, even claims a "right to exist" (S5) for individual spiders and demands the same fine for killing a spider as for killing a cat or dog, "since they all are living beings." On these grounds, W1 and C5 explicitly call for employing the Kantian imperative as the golden rule of conduct towards not only human, but also non-human beings.

We found intriguing social dynamics regarding participants' stance to humans' place in nature. Participants critical of the model animals attributed a state of being alienated from nature to people favorable them and vice versa. Critical participants depicted the favorable party as aloof "urban residents" naively endorsing and "romanticizing" (W7), e.g., wolf recovery, and "lacking in knowledge" (W2) about nature, as well as in personal experience with wildlife in general. Critical participants purported that favorable people base their judgment on an excessively "sensitive" (W2) and "emotional" (C3, W7) biocentric view that one-sidedly focuses on only the one "darling" (C3) species whilst neglecting the systemic whole of an ecosystem. Assumptions of urban populations being more positive towards potential problem wildlife mirrors the current state of knowledge in human dimensions research [6,54,114]. Interestingly, however, the very same attributions were raised by wildlife-favoring participants against critical parties. For example, it was raised that people who, e.g., hold critical views of corvids preying on small game, may not be educated enough and too "soft" (C2) to accept the fact that "nature can be cruel" (C4). Additionally, some favorable participants expressed the idea that critical people are and seek to be so "separated" (C2, C5) from natural processes and "niminy-piminy" (S2) to a degree that they will not tolerate wild animals in their proximity. Participants favorable

to the model animals moreover portrayed critical people as being "ill-natured" (W1) and "coldhearted" (C5). Favorable participants assumed that critical people pursue a quest for "subjugating" (C5) nature because they generally repudiate "all sorts" (C2) of wildlife and seek to "distinguish themselves" as having a higher status (C5) because they are "weak of character" (S5). In this vein, W1 put forward that wolf opponents "orgasm when shooting wolves".

### 3.3.2. The Question of Control

The theme of "control" is pertinent in the context of human–wildlife interactions. It is mostly used in the sense of lethally controlling population densities. Yet, rather than being just a tool of wildlife management, our results show that "controlling" wildlife is multifaceted and goes much deeper. People seek and fear losing control in direct encounters with individual wild animals (cf. [115,116]) and also with regard to large-scale changes such as resurging wolf populations. Moreover, we find that two conditions of (fear of) losing control need to be distinguished: an acute and a latent form. Again, participants showed starkly opposing stances with regard to seeking control over wildlife.

Latently, the concept of control is triggered by all three model animals because many participants are uncertain about how to interact with them: Wolves', corvids', and spiders' behaviors are said to be difficult to predict and influence (cf. [117]). This uncertainty seems to cause an unsettling sense of omnipresent threat that appears to be an undercurrent in participants' way of thinking about those animals. Correspondingly, in the larger scope of human relations to wolves, corvids, and spiders, their ostensible agency seems to provoke some participants' deep-seated fears of being forced to surrender control to overpowering events (cf. [118]). Such a situation of feeling overpowered could be constituted, for example, by the potentially unbearably "horrendous noise level" (C6) caused by rooks progressively moving into city parks and purportedly impeding their human neighbors from finding sleep. It is against the backdrop of such visions of being foreseeably overpowered and bereft of control on the small and large scales of interacting with wildlife that participants demand a curbing of wolf and corvid numbers before they become "excessive" (W5; cf. [11]). Analogously, participants preventively remove spiders from their homes to avoid "overstepping the line of nature propagating uncontrollably in my shelter" (S6). Even participants claiming to be tolerant of potentially problematic wildlife often add that their tolerance be conditional with regard to wildlife numbers staying below a certain, often unspecified, limit. Wildlife numbers for these participants, seem to be a means of expressing the idea of a tipping point which marks an amorphous demarcation between maintaining control and "things getting out of hand" (W5).

Conversely, other participants rejoice in both situational displays of animal agency and in the fact that generally, humans may not be the only beings deliberately shaping their environments, but co-create them with animal agents [8,9,11]. These participants attribute themselves and wildlife-friendly people with "equanimity" (S2) and have "no interest whatsoever to interfere" (W1) with wild animals who "ought to live like they themselves choose to live" (S3). This attitude relates to the connivance mentioned in the previous section of rejecting a potential claim of humans to master and manage nature. Participants who are happy to cede control over wildlife hold that generally, "appreciating nature includes allowing her to sort things out on her own" (C2), and that in particular, "live and let live, that is: acting out of respect for other beings" (S5) ought to be the guiding principle for interacting with animals, including potentially problematic wildlife (cf. [66]). Accordingly, these participants, such as C5, scold critical peoples' attempts to "subdue everything! They don't allow any freedom for the animal." C5 and others accept potential nuisances, e.g., corvids' littering or croaking because they see it as a necessary facet of life: "That is not 'too loud' or anything, that is just being alive!"

However, even for participants accepting or supportive of animal agency, a general stance of allowing may be overturned in case of an acute loss of control. In Section 3.1, we portrayed how C4's general eco-centric conviction gave way to pungent wrath towards

ravens menacing her lambs when she was acutely affected by the unbridled behavior of wildlife which she said to have cherished previously. It seems that when people are acutely affected, a sudden and seemingly mechanistic struggle for regaining control over threatening events sets in during which even thorough values may be put out of operation temporarily, or selectively, with regard to the particular wildlife species. The case of W6 provides another example. As with C4, he holds strong ecological values and goes to great lengths in providing a rich habitat for all species on his meadows. Yet, he selectively exempts wolves and magpies from his otherwise apparently seamless attitude of connectedness of humans and nature, literally demanding that wolves be "eradicated" from Central Europe and magpies be lethally controlled. The stark contrast between W6′s general conviction and his stance towards these two species seems to be due to W6′s controlling personality style (cf. [119]) that became apparent in multiple ways throughout the interview. Wolves and magpies—who allegedly harassed a wounded mare—not only menace W6′s horses which he considers "part of the family", but also symbolically threaten his position as a patriarch holding control not only over his human and equine family, but also over all life unfolding on his property. As long as animals' behavior aligns with his needs, W6 presents as a gracious ruler. Once animals' agency may be understood as running counter to his authority, W6 reflexively seeks to subdue the seemingly disrespectful beings. Concurrently, research has shown that people with authoritarian values support "restricting the free movement of a new animal species in the wild" if they "feel threatened" by that animal [62] (pp. 804, 812). However, participants with much less rigid personality styles reported states of acute helplessness akin to "freeze-mode" (S4) when feeling threatened by an animal whose behavior they cannot control. Arachnophobe and vegan S1 shared his feeling of "guilt" when killing a spider as a means of "just getting rid" of the threat, because doing so blatantly contradicts his ethos of leading a cruelty-free life [70]. Yet, the response of lashing about in a state of acute helplessness is not limited to panic caused by animal phobia. As the case of C4 evidences, seemingly overshooting self-defense also can be triggered by a sense of feeling "abandoned" and left to fend for oneself. As it seems, such a state of helplessness and abandonment can be triggered by different kinds of wildlife when they seem to menace corporeal or ideational elements integral to a person's identity, thus raising a counter-aggression which checkmates rational thought and general interpretation patterns.

### 3.3.3. Further Questions and the Interlocking of Layers: Future Research

Viewed from the layer of overarching factors, wolves, corvids, and spiders are not only wild animals, but carriers of human projections and keys that unlock meanings seated deep within an individual's mind. In addition to the question of humans' place in nature and of how to deal with the uncontrollability of wildlife, there are further questions raised by wild animals acting as triggers. For example, people have a general penchant for feeling personally targeted by wild animals' behavior [70]. Additionally, we have found that symbolic associations to corporeal and metaphoric darkness are attached to wolves, corvids, and spiders. Corporeal darkness is evoked, since wolves are said to be "nocturnal" (W2, W3), most well-known corvid species bear black feathers, and spiders lurk in dark corners and are prototypally represented as "black" (S4). Associations to metaphoric darkness, i.e., questions of how to conceive of evil and how to deal with the shadow aspects within one's own mind [2] are evident, e.g., in the rich application of moralizing terminology with regard to animals' behavior (see Section 3.2, and [4]). This underscores how issues of good and evil are being negotiated in actual or conceptual interactions of humans with wildlife (cf. [76,79]).

Moreover, associations to darkness in the sense of a Jungian shadow—as the unwanted and thus disowned and repressed aspects of one's own personality—took idiosyncratic forms for participants. For example, S5 divulged how his penchant for aggressive animals, such as tarantulas, was a means to develop an "armor" shielding his "big heart" against the perpetrations he has endured by "reckless" people. Likewise, S3 self-reflectively discovered

during the interview how his wish to provide "freedom" for his pet spiders was a way to vicariously heal his own want for aggressive self-assertion from the trauma of being constrained by his intrusive mother. In other cases, participants were unaware of potential projections onto the model animals that, we assume, resulted from dynamics related to ongoing repression of the shadow. W6's stark repudiation of wolves (see Section 3.3.2) is an illustrative example. We recommend purposefully targeting deep psychological hypotheses such as these as a fecund avenue for future research.

These examples illustrate how individual persons' biographies predispose them to responding in their own specific way to an overarching question (cf. [100]), and how conversely, overarching patterns emerge from individual persons' idiosyncratic views. This becomes apparent also by the systematic mapping of participants' life themes and utopian visions (see Table 1). For example, those participants whose life theme is a struggle against a presumed cold-heartedness of other people with regard to animals or to themselves, tend to hold a vision of harmony within human community or even among all living beings. Accordingly, they will be inclined to eco- or biocentric values, and to welcoming animal agency, personhood, and recalcitrancy against human dominance. Conversely, if a person devotes their vocation to the conviction that humankind is the pivotal manager in a natural system, their vision will conform to a wise-use ideal, characterized by anthropocentric value orientations and will have little interest in ceding control to non-human agents or self-organizing natural phenomena [101,120]. As the focus of this qualitative study is on describing how the factors we have noted feed into the overarching themes for a limited number of subjects, we cannot systematically assess and statistically support those mappings. Kellert's [121] types of wildlife value orientations provide such large-scale assessments, and Jürgens et al. [72] provide both a detailed analysis of, and quantitative evidence for a mapping between general themes and views of wildlife for the case of human–wolf relations. Comparing the presumed systematicity of person-specific stances to trans-personal patterns in various cases of human–wildlife relations is another future direction for research building on our findings.

Our proposal for how factors of the three layers interact concurs with the key assumption captured in Kansky et al.'s [28] wildlife tolerance model: namely that personal costs and benefits, as well as prior contact with a wildlife species—i.e., person-specific factors—mediate the formation of an individual's response towards that animal. Moreover, many of the variables comprised in the model's inner tier—e.g., wildlife value orientations, anthropomorphism, and perceived behavioral control—correspond to the themes of the overarching questions presented here. Moreover, our three-layer framework of impact factors in human–wildlife relations is open to accommodating further factors and processes in addition to the ones discussed in this study. For example, while most variables listed in extant synopses (see Section 1.1) correspond to concepts captured within our model, others, such as cultural character [27], preventive measures [26], or media, law and policy [17], that we have not covered due to the particular focus of our investigation, can still be straightforwardly incorporated into our framework as person-specific, species-specific, and overarching factors, respectively. Appendix A Table A1 lists many further pertinent concepts and how they can be integrated into the layers of the framework presented here.

While our framework has been conceived based on data derived from a purely German-speaking sample, the seamless way in which our findings are congruent with and complement extant results to form a comprehensive picture of the human dimensions suggests that our results are of general relevance for explaining the intense and polarized emotions that habitually rage in all kinds of human–wildlife encounters. The layer of overarching factors harbors thorough questions about the human condition in general and the human relationship to nature that are raised by wildlife in species-specific ways. These questions go straight to the heart of people's self-image as individuals and as parts of humankind. Additionally, on a more practical level, they are entwined with issues—e.g., finding sleep, safe-guarding human and non-human family members, or pursuing one's vocation—that matter to people and directly affect their livelihoods in some instances. Therefore, those

questions touch on issues of existential relevance, hence the emotional vigor when people encounter wildlife that force them to address those issues. Depending on the particular perspectives shaped through their biographies, people give person-specific answers to those questions, hence the polarized stances we observe in human–wildlife interactions.

Future research may apply these insights to analyzing various cases of human relations to wildlife. Additionally, the more precise findings about the overlapping and uniquely defining characteristics of people's representations of wolves, corvids, and spiders may be transferred to assessing people's mental images of other kinds of wildlife where similar patterns exist. For instance, factors contributing to people's views of wolves may help elucidate human relations to bears [31,54], large feline predators [17], elephants [122], or other charismatic megafauna [123] who also may challenge people's lives or livelihoods. Perspectives on corvids may shed light on human responses to wild boars [124] or beavers [25] who do not existentially threaten Central European humans but are considered significant nuisances by some and greeted as epitomes of natural resilience by others. Human relations to spiders may mirror and help to understand why many insect species, mice, snakes, and bats who are exhibiting similar physical features or behaviors as spiders trigger intense reactions in people [93,125–127]. Future comparative research may further elucidate the scope of shared themes and unique features from which a person's mental image of an animal may be composed and that jointly define the layer of species-specific factors in human–wildlife relations.

In summation, human responses to ecologically disparate kinds of wild animals share overarching impact factors and can be understood according to a common three-layer structure. This knowledge can be directly applied to analyzing various kinds of human–wildlife relations and may moreover be used to leverage conservation and management of wildlife populations. In particular, our results suggest that management approaches can be transferred between seemingly different cases and only need to be applied to the person- and species-specific circumstances of any given situation. Jürgens and Hackett [2] make concrete suggestions for approaches to practical wildlife management that are based on the overarching questions elucidated here.

## 4. Conclusions

In accordance with previous proposals [17,19,20,26–28], our qualitative study confirms that there is a multitude of variables and processes contributing to human–wildlife relations. Yet, to our knowledge, we provide the first empirical support for the idea that those variables and processes apply to vastly different taxa of wildlife. Integrating extant conceptions and corroborating the approach introduced in Jürgens and Hackett [2], we propose a three-layer framework of person-specific, species-specific, and overarching factors. We explain how these layers interplay in bringing forth the well-known intense and polarized reactions of people to potentially problematic wildlife: Regarding the layer of overarching factors, the comparative investigation of wolves, corvids, and spiders suggests that encounters with virtually all kinds of wildlife raise the same deep questions, i.e., about the place of humans in nature, about whether and how to exert control in the face of non-human agency, and about how to respond to symbolic associations activated by the animals. Simultaneously, responses to wildlife are highly personal since the layer of person-specific factors causes people to choose individual answers to those overarching questions. Finally, the intermediate layer of species-specific factors explains why some animals are more prone than others to elicit human–animal conflicts and elucidates the particular character, i.e., the specific challenges and affective tone, of interactions with a given kind of animal.

By showing how this three-layer framework can be applied to wildlife as ecologically diverse as a mammalian predator, an avian hemerophile, and an invertebrate, the findings of this study broaden the existing conceptions for shared factors in different human–wildlife interactions. By elucidating the internal structure of these factors as well as their interplay, we underscore that human dimensions are a significant parameter in wildlife ecology

and evolution in the Anthropocene. Understanding that this parameter is not monolithic yet exhibits consistent mechanisms that systematically shape human relations to diverse kinds of wildlife empowers practitioners and decision makers to effectively leverage the conservation and the management of wild animals and of the people coexisting with them.

**Author Contributions:** Conceptualization, U.M.J., P.M.W.H., M.H. and A.P.; methodology, U.M.J., P.M.W.H. and M.H.; validation, P.M.W.H. and M.H.; formal analysis, U.M.J.; investigation, U.M.J.; resources: M.H.; data curation: U.M.J., P.M.W.H. and M.H.; writing—original draft preparation, U.M.J.; writing—review and editing, U.M.J., P.M.W.H. and M.H.; visualization, U.M.J.; supervision, P.M.W.H., M.H. and A.P.; project administration: A.P.; funding acquisition, U.M.J. and M.H. All authors have read and agreed to the published version of the manuscript.

**Funding:** The first author is grateful to the Deutsche Wildtier Stiftung who has supported the project through their Research Prize 2015. Moreover, this research has been supported by funds of the Eidgenössische Forschungsanstalt für Wald, Schnee und Landschaft WSL who also cover the APC. The authors declare that they have no known competing financial interests or personal relationships that could have appeared to influence the work reported in this paper.

**Institutional Review Board Statement:** Ethical review and approval were waived for this study. Prior to conducting the research, UMJ presented her paradigm to the ETH ethics committee and was grated consent for pursuing this plan by the responsible agent without a need to apply for a formal approval by either the ETH ethics committee or the cantonal ethics committee in Zürich.

**Informed Consent Statement:** Informed consent was obtained from all subjects involved in the study.

**Data Availability Statement:** Copies of interview transcripts (in German) can be made available upon request.

**Acknowledgments:** In heartfelt gratitude, the first author is indebted to her late supervisor Werner Suter for his enthusiasm about and support for this project from hour one. Moreover, she would like to thank Margarita Grinko for her suggestions that greatly improved an earlier version of the manuscript. Many thanks also go to Athina Nalbanti for graciously providing technical equipment for conducting the interviews.

**Conflicts of Interest:** The authors declare no conflict of interest. The funding institutions had no role in the design, execution, interpretation, or writing of the study.

## Appendix A

**Table A1.** Corresponding levels and concepts found to impact human–wildlife relations in extant models and within the three-layer framework of this study.

| *Three Layers* | *Factors Found to Impact Human–Wildlife Relations in This Study* | König et al., 2020 [26] | Manfredo and Dayer, 2004 [27] | Dickmann, 2010 [20] | Bathia et al., 2020: 5 Ultimate Factors [17] | Kansky and Knight, 2014 [19] | Kansky, Kidd and Knight, 2016: Wildlife Tolerance Model [28] | Pertinent Concepts in the Literature |
|---|---|---|---|---|---|---|---|---|
| *Person-specific factors - individual identity* | | | | micro-level | | Sociodemographic variables | | |
| | *personal affectedness and acuteness of being affected* | | micro-level | perception of risk: actual and perceived costs | perception of risk | tangible and intangible costs tangible and intangible benefits | tangible and intangible costs tangible and intangible benefits | |
| | | | | social risk factor: vulnerability and wealth environmental risk factor: land use management | resource dependence, e.g., wealth, occupation, education | land use and dependence, e.g., wealth | | |
| | | context adaptation on the local level | | social risk factor | | exposure and experience | exposure and positive/negative meaningful experiences with species | closeness to established wildlife populations [12] |
| | *Interpretation patterns [29] life themes and visions originating in a person's biography personality traits* | | micro-level: behavior | social risk factor | | | personal habit | |
| | | | micro-level: norms and attitudes | social risk factor: beliefs and values | | attitudes towards species | general values personal and social norms | sociocultural value concept [81] |
| | | | micro-level: cognition and affect | | | salience of animal knowledge | interest in animals empathy | |
| *person-specific factors - collective identity* | *ascriptions to the opposing party:* <br> - *assumed incompetence* <br> - *assumed alienation from nature* | | | social risk factor: distrust and animosity | social interactions | cohort and demographic group | | membership in stakeholder group |
| | | | macro-level: cultural character | social risk factors: religious beliefs | | | | religion animism [128] |

Table A1. *Cont.*

| Three Layers | Factors Found to Impact Human–Wildlife Relations in This Study | König et al., 2020 [26] | Manfredo and Dayer, 2004 [27] | Dickmann, 2010 [20] | Bathia et al., 2020: 5 Ultimate Factors [17] | Kansky and Knight, 2014 [19] | Kansky, Kidd and Knight, 2016: Wildlife Tolerance Model [28] | Pertinent Concepts in the Literature |
|---|---|---|---|---|---|---|---|---|
| | | capacity building and damage prevention on the regional and local levels | | level of wildlife damage environmental risk factor: behavior and management of species; physical features of environment | nature of interaction with the animal, e.g., frequency and magnitude of conflict | (perceived) species characteristics, e.g., abundance and population density mitigation measures | | a species' ecology |
| *Species-specific factors* | *mental image of the animal: perceived features of, as well as beliefs and stereotypes about a specific animal species that are shared between participants (features including, but transcending the species' ecology)* | | micro-level: affect and cognition | | | | taxonomic bias anthropomorphism | factors shaping species preference [97,98] stereotype content model [129] Big Bad Wolf stereotype [4] Anthropomorphism [130,131] mind perception [132] species' belonging to a landscape [6,81,82] |
| | | | micro-level: affect | | affective dimension of risk perception | intangible costs: psychological costs of danger or risk intangible benefits: positive emotions | intangible costs: negative emotions, fear, danger, nuisance and stress intangible benefits: positive emotions positive and negative meaningful events | affect for the species [133] species-specific patterns of fear [115,134–136] |
| *Overarching, fundamental questions raised by all human–wildlife interactions* | *A competition for resources exists between humans and wildlife—how should it be resolved? What is a "fair" balance between humans' and animals' needs?* | governance and legal frameworks on international to regional to local level | macro-level | social risk factors: human–human conflicts; inequality and power | perception of risk: media, and law and policy | legal status of land landscape characteristics property characteristics | trust in institutions | political geographies politicization of conflict [137] urban–rural divide NIMBY-effect [138] |

**Table A1.** *Cont.*

| *Three Layers* | *Factors Found to Impact Human–Wildlife Relations in This Study* | König et al., 2020 [26] | Manfredo and Dayer, 2004 [27] | Dickmann, 2010 [20] | Bathia et al., 2020: 5 Ultimate Factors [17] | Kansky and Knight, 2014 [19] | Kansky, Kidd and Knight, 2016: Wildlife Tolerance Model [28] | **Pertinent Concepts in the Literature** |
|---|---|---|---|---|---|---|---|---|
| | *The place of humans in nature:*<br>*Are humans . . .*<br>*- . . . the centerpiece of the world (anthropocentrism)*<br>*or*<br>*a curse for the remainder of creation*<br>*(anti-anthropocentrism)*<br>*or*<br>*individual beings amongst individual beings*<br>*(biocentrism)*<br>*or*<br>*one species in a web of species (ecocentrism)?*<br>*- . . . connected with nature*<br>*or*<br>*distinct from nature?*<br>*- . . . endowed with a responsibility to care for their fellow animals*<br>*or*<br>*endowed with the right to manage nature?*<br>*Are wild animals to be viewed as*<br>*- collections of individuals*<br>*or*<br>*- as the whole of a species?* | | macro level: Wildlife value orientations "mutualism and domination" | | value orientations | | wildlife value orientations | Kellert's [121] ten types of value orientations and two fundamental dimensions "utility" and "affect" anthropocentrism vs. biocentrism vs. ecocentrism, and pluralism [102,107] biophilia [90,139] value basis for environmental concern [103] new environmental paradigm [104] separation vs. coexistence model in conservation [109]<br><br>perspectives of hyper-separation vs. collaboration [140] dualistic vs. biocultural view of wilderness [141] |
| | *Control:*<br>*Dealing with wildlife agency:*<br>*- allowing free reign*<br>*or*<br>*- restricting wildlife behavior?*<br>*Reacting to acute affectedness:*<br>*- helplessness*<br>*and/or*<br>*- reactive aggression?* | | micro-level: perceptions of control | | | | self-efficacy behavioral control | control one's own response [115] desirability of control [119] locus of control [142] control in terror management Theory [143] autonomy of nature [118] |

**Table A1.** *Cont.*

| *Three Layers* | *Factors Found to Impact Human–Wildlife Relations in This Study* | König et al., 2020 [26] | Manfredo and Dayer, 2004 [27] | Dickmann, 2010 [20] | Bathia et al., 2020: 5 Ultimate Factors [17] | Kansky and Knight, 2014 [19] | Kansky, Kidd and Knight, 2016: Wildlife Tolerance Model [28] | Pertinent Concepts in the Literature |
|---|---|---|---|---|---|---|---|---|
| | *symbolic meaning associated to wild animals e.g., associations to "darkness" (evil, mortality); expressed through a prototypical dark exterior* | | | symbolism | | | | landscape as symbolic environment [100] deeper levels of conflict [101] terror management theory [144,145] |

**Table A2.** Sociodemographic information on interview participants.

| Code | Collective Identity—Stakeholder Group | | | | | | | | Nationality | Gender | Age | Degree and Manner of Being Affected | Attitude to Model Wildlife |
|---|---|---|---|---|---|---|---|---|---|---|---|---|---|
| | H | O | Animals owned | P | L | E | A | S | | | | | |
| W1 | | | | | L | | A | S | G | M | 59 | no wolf area, urban resident | positive |
| W2 | H | | | | | | | | G | M | 73 | lives in area of dispersing wolves | ambivalent |
| W3 | H | | | | | | | | G | M | 70 | lives in area of dispersing wolves | ambivalent |
| W4 | H | | | | | | | | G | M | 73 | lives in area of dispersing wolves | negative |
| W5 | H | | | | | | | | G | M | 70 | lives in area of dispersing wolves | neutral |
| W6 | | O | Horses | | | E | | | G | M | 83 | lives in wolf area; unconfirmed wolf attack on horses | negative |
| W7 | H | | | P | L | | | | G | M | 50+ | lives in wolf area; lobbies for affected farmers | ambivalent |
| C1 | | | | | | E | | S | CH | F | 33 | no corvid populations nearby | neutral |
| C2 | H | | | | | | | | G | M | 43 | lives close to rookery | positive |
| C3 | H | | | | L | | | | G | M | 47 | avid hunter of crows | positive |
| C4 | | O | Sheep | | | E | | | G | F | 60 | alleged attack on lambs | ambivalent |
| C5 | | | | | | | A | | G | F | 59 | lives close to rookery | positive |
| C6 | | | | | | | | | G | M + F | 70 | live close to rookery | negative |

**Table A2.** *Cont.*

| Code | Collective Identity—Stakeholder Group | | | | | | | | Nationality | Gender | Age | Degree and Manner of Being Affected | Attitude to Model Wildlife |
|---|---|---|---|---|---|---|---|---|---|---|---|---|---|
| | H | O | Animals owned | P | L | E | A | S | | | | | |
| S1 | | | | | | | A | | G | M | 27 | phobic | negative |
| S2 | | | | | | | A | | G | F | 33 | normal level of affectedness | neutral |
| S3 | | | Pet spiders | | | | | | G | M | 27 | owns tarantulas | positive |
| S4 | | | | | | | | | G | F | 29 | phobic | negative |
| S5 | | | Pet spiders | | | | | | G | M | 30 | owns tarantulas | positive |
| S6 | | | | | | | A | | G | M | 30 | previously phobic | ambivalent |
| S7 | | | Pet spiders | | | | | S | G | M | 36 | researches spiders | positive |
| **total** | **7** | **2** | **-** | **1** | **3** | **3** | **5** | **3** | **19 G** **1 CH** | **14 M** **5 F** **1 Couple** | **Md: 47** **Mn: 50** | **-** | **-** |

H, hunter; F, livestock Owner; P, (former) politician; L, political lobbyist for their respective cause; E, active environmentalist; A, animal welfare/rights activist; S, scientist; G, German; CH, Swiss. M, male; F, Female; Md, Median; Mn, Mean; n.a., not applicable.

**Appendix B**

Questions and Prompts used in the in depth interviews. The interview guidelines have been originally used in German and are presented here in a translated version.

1.  Have you had any personal experiences with wolves/crows/spiders?
2.  Which feelings arise when you think about wolves/crows/spiders?
3.  What is it about wolves/crows/spiders that evokes these thoughts and feelings?
4.  (Arrangement of figures:) How would you describe your personal relation to wolves/ crows/spiders? Can you illustrate your relation to wolves/crows/spiders with these figures?
5.  Could wolves/crows/spiders stand as symbols for something? If so, for what?
6.  Other people might see wolves/crows/spiders in a different light. What distinguishes you from these people? Why do you like/dislike wolves/crows/spiders while others dislike/like them?
7.  (Arrangement of figures:) How do you think would these people that like/dislike wolves/crows/spiders arrange these figures to depict their view of wolves/crows/spiders?
8.  There are many different opinions about whether humans should restrict their freedom in order to be considerate of wildlife. What do you think?
9.  What enrages you about other people's behavior towards wolves/crows/spiders?
10. How would you explain to a child what is key in human-wolf/crow/spider relations?
11. If you were granted three wishes with regard to wolves/crows/spiders—what would they be?
12. Imagine you were a god/goddess who could arrange the world in any possible way. You could change and create everything: Humans, animals, landscapes—just everything. How would you arrange the world in a way that human–wildlife conflicts are eliminated?
13. Ideally, what ought to be the role of humans in nature?
14. What constitutes the biggest challenge in human coexistence with wolves/crows/spiders?
15. What is the biggest possibility inherent in human coexistence with wolves/crows/spiders?
16. If you had the power to decide: What would be a realistic solution to the conflict between humans and wolves/crows/spiders?

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
