# Peer review of "Wolves, Crows, Spiders, and People: A Qualitative Study Yielding a Three-Layer Framework for Understanding Human–Wildlife Relations"

_diversity, doi:10.3390/d14080591_

Round 1

Reviewer 1 Report

The study, reported in the manuscript, is a fascinating insight into human-wildlife relations. The sample of informants is small and biased towards central Europeans of one language group. The authors argue that the sample was exhaustive regarding the aims of the study and the reader is required to put some faith into that argument. Even so the authors recognise the limitations of the sample in the Discussion/Conclusions and advocate further research to justify the generality of their findings. They present a framework to structure such research and its interpretation.

The manuscript is well-written, but I have a few suggestions given in the following for improvement.

The introduction is expansive, and I found Table 1 cumbersome and difficult to follow. Furthermore, Table 1 seemed to pre-empt the conclusions by reference to the three-layer framework, an aim and outcome to the study. I was more informed by the text and suggest removing Table 1 or at least making it more readable (e.g. breaking into three tables under each separate layer and perhaps distinguishing macro and micro-layers by some emphasis such as italics or bold rather than confusing headings within a column). Even if Table 1 is modified then it needs to be justified as part of the Introduction rather than an appendix or conclusion.

The results are presented in large tables but unlike Table 1, they are more comprehensible and as such are justified.

The discussion is quite expansive and introduces many nuances to the results. I found this informative and interesting, and the length justified.

Minor corrections:

Line 50, factor at the expense of

Line 67 and section 3.2 and following – suggest critical vs favourable rather than favoring

Line 126, they do not

Line 147, unclear what ‘development’ of wildlife populations means

Line 166, Twenty human

Line 206,  Participants

Line 426, experience of the hunt, Wolves are not good managers, Corvids behave impudently

Line 434, If one takes

Line 621-2, place second clause beginning (sic) ‘Dependent’ last

Line 766, presents as

Line 807, targeting deep

Line 869, who do not

Author Response

Dear Reviewer,

we are delighted to receive your positive, friendly and amazingly detailed feedback. Thank you very much for even providing line-specific suggestions! We have incorporated all of them into the revised version of the manuscript.

In the following, we respond to the two major comments and explain how we implemented them.

You said: “The introduction is expansive, and I found Table 1 cumbersome and difficult to follow. Furthermore, Table 1 seemed to pre-empt the conclusions by reference to the three-layer framework, an aim and outcome to the study. I was more informed by the text and suggest removing Table 1 or at least making it more readable (e.g. breaking into three tables under each separate layer and perhaps distinguishing macro and micro-layers by some emphasis such as italics or bold rather than confusing headings within a column). Even if Table 1 is modified then it needs to be justified as part of the Introduction rather than an appendix or conclusion.

 --> Thank you very much for this suggestion. Yes, indeed, we felt that table 1 was quite a handful if included as part of the main body of the manuscript. We can also see how its structure preempts the results and conclusions. Therefore, we followed your recommendation to include it as an appendix instead. We adapted the text accordingly.

You said: “Line 147, unclear what ‘development’ of wildlife populations means”

--> We have deleted the term "development", as the sentence is actually better digestible and more precise without it.

Reviewer 2 Report

The authors proposed and used a three-layer framework of person-specific, species-specific, and overarching factors to explain human-wildlife relations. They applied it to such different taxa as a mammal, a bird and an invertebrate. Their ideas, applications and analyses are both novel and interesting, furthering the field of human dimensions of wildlife management.

 This is a manuscript of high quality, worth of publication in Diversity.

Author Response

Dear Reviewer,

We would like to thank you very much for taking the time for reviewing our manuscript! We are profoundly delighted that you provided favorable feedback on our work and consider it a valuable contribution to the field.

We have improved the manuscript in accordance with the minor changes recommended by another reviewer, but are confident that you will find the revised version as adequate as the original draft.